# Potential hidden *Plasmodium vivax* malaria reservoirs from low parasitemia Duffy-negative Ethiopians: Molecular evidence

Abnet Abebe[1,2]*, Isabelle Bouyssou[3,4], Solenne Mabilotte[5], Sisay Dugassa[1], Ashenafi Assefa[2,6], Jonathan J. Juliano[6], Eugenia Lo[7], Didier Menard[3,5,8,9], Lemu Golassa[1]*

**1** Aklilu Lemma Institute of Pathobiology, Addis Ababa University, Ethiopia, **2** Ethiopian Public Health Institute, Addis Ababa, Ethiopia, **3** Institut Pasteur, Université Paris Cité, Malaria Genetics and Resistance Unit, INSERM U1201, F-75015 Paris, France, **4** Sorbonne Université, Collège Doctoral ED 515 Complexité du Vivant, Paris, France, **5** Université de Strasbourg, Institute of Parasitology and Tropical Diseases, Dynamics of Host-Pathogen Interactions, F-67000 Strasbourg, France, **6** Institute of Infectious Disease and Global Health, University of North Carolina at Chapel Hill, Chapel Hill, North Carolina, United States of America, **7** Department of Biological Sciences, Bioinformatics Research Center, University of North Carolina at Charlotte, United States of America, **8** Institut Pasteur, Université Paris Cité, Malaria Parasite Biology and Vaccines Unit, Paris, France, **9** CHU Strasbourg, Laboratory of Parasitology and Medical Mycology, Strasbourg, France

\* abnetabas@gmail.com (AA); lgolassa@gmail.com (LG)

**Data Availability Statement:** All relevant data are within the manuscript.

## Abstract

### Background

The interaction between the *Plasmodium vivax* Duffy-binding protein and the corresponding Duffy Antigen Receptor for Chemokines (DARC) is primarily responsible for the invasion of reticulocytes by *P. vivax*. The Duffy-negative host phenotype, highly prevalent in sub-Saharan Africa, is caused by a single point mutation in the GATA-1 transcription factor binding site of the DARC gene promoter. The aim of this study was to assess the Duffy status of patients with *P. vivax* infection from different study sites in Ethiopia.

### Methods

A cross-sectional study was conducted from February 2021 to September 2022 at five varying eco-epidemiological malaria endemic sites in Ethiopia. Outpatients who were diagnosed with *P. vivax* infection (pure and mixed *P. vivax/P. falciparum*) by microscopy and Rapid Diagnostic Test (RDT) were subjected to PCR genotyping at the DARC promoter. The associations between *P. vivax* infection, host genotypes and other factors were evaluated.

### Result

In total, 361 patients with *P. vivax* infection were included in the study. Patients with pure *P. vivax* infections accounted for 89.8% (324/361), while the remaining 10.2% (37/361) had mixed *P. vivax/P. falciparum* infections. About 95.6% (345/361) of the participants were Duffy-positives (21.2% homozygous and 78.8%, heterozygous) and 4.4% (16/361) were Duffy-negatives. The mean asexual parasite density in homozygous and heterozygous

**Funding:** The author(s) received no specific funding for this work.

**Competing interests:** The authors have declared that no competing interests exist.

Duffy-positives was 12,165 p/µl (IQR25-75: 1,640–24,234 p/µl) and11,655 p/µl (IQR25-75: 1,676–14,065 p/µl), respectively, significantly higher than that in Duffy-negatives (1,227p/µl; IQR25-75: 539–1,732p/µl).

## Conclusion

This study confirms that Duffy-negativity does not provide complete protection against *P. vivax* infection. The development of *P. vivax*-specific elimination strategies, including alternative antimalarial vaccines should be facilitated by a better understanding of the epidemiological landscape of *vivax* malaria in Africa. More importantly, low parasitemia associated with *P. vivax* infections in Duffy-negative patients may represent hidden reservoirs of transmission in Ethiopia.

### Author summary

*Plasmodium vivax* generally receives less attention than *P. falciparum* and was neglected in sub-Saharan Africa. However, the characteristics of *P. vivax* infection in Duffy-negative individuals, and the distribution of Duffy blood group in different eco-epidemiological zones and ethnic groups of Ethiopia are not well documented. Here, we determined the Duffy genotypes of *P. vivax* infected patients across broad regions of Ethiopia. It is clear that Duffy negative individuals in Ethiopia are not fully protected against *P. vivax* infection, and that these infections in Duffy negatives are often associated with low parasitemia. Our findings lend support to the notion that *P. vivax* may have developed a Duffy-independent erythrocyte invasion pathway and/or evolution in host immune responses.

## Introduction

In 2021, there were an estimated 247 million cases and 619,000 deaths from malaria worldwide. Of these, 2% of the cases were due to *P. vivax*, mainly in the Horn of Africa, South America, India and Asia [1]. *P. vivax* malaria is recognized as a cause of severe morbidity and mortality, with a significant negative impact on health in endemic countries [2–4]. The unique biological features of *P. vivax*, such as hypnozoite formation and early gametocytogenesis, pose a challenge to global malaria elimination efforts [5].

The *P. vivax* invasion pathway into reticulocytes was previously thought to be primarily dependent on the interaction between the Duffy binding protein (PvDBP) and its cognate receptor, the Duffy Antigen Receptor for Chemokines (*DARC*) [6–8]. Polymorphisms in the promoter of the *DARC* gene alter the expression of the Duffy antigen protein and determine an individual's Duffy blood group [9]. A point mutation in the GATA-1 transcription factor binding site of the promoter of the gene at position−67 encodes the change of a T nucleotide to a C nucleotide and underlies the Duffy-negative (erythrocyte silent) phenotype [10–12]. Individuals with this mutation do not express the Duffy antigen on the surface of their erythrocytes [13].

Duffy-negative phenotype in the population occurs at low frequency in Caucasians (~3.5%) [14], and has a prevalence of almost 100% in West Africa, > 80% in African Americans [15], 20–36% in East Africa and 84% in southern Africa [16]. In Ethiopia, a study reported 94/416

(~23%) of the febrile patients from the six health centres/hospitals were Duffy-negatives, though only two of them were confirmed with *P. vivax* [17].

The scientific paradigm that *P. vivax* exclusively invades Duffy-positive reticulocytes through the interactions of PvDBP and *DARC* was established, and it was assumed that *vivax* malaria was rarely transmitted or even absent in regions where Duffy-negativity predominates, such as sub-Saharan Africa [18,19]. However, recent studies have shown that *P. vivax* can infect Duffy-negative reticulocytes [20], and is no longer a barrier to such infections [21]. In East Africa, the prevalence of *P. vivax* in Duffy-negative populations ranges from 3% [22] to ~12% in Ethiopia [16] and 3% [16] to ~18% in Sudan [9]. Although several studies have demonstrated the presence of *P. vivax* infection in Duffy-negative individuals, the distribution of these infections in different eco-epidemiological zones and ethnic groups are not well documented, especially in Ethiopia. This study compares the prevalence of Duffy blood group genotypes of *P. vivax* infected patients among different eco-epidemiological zones of Ethiopia.

# Materials and methods

## Ethical statement

Ethical approval was obtained from Aklilu Lema Institute of Pathobiology Institutional Review Board (IRB) (Ref No: ALIPB IRB/34/2013/20), Addis Ababa University, and ministry of education (MoE) via its national research ethical review committee (Ref No:7/2-514/m259/35), Addis Ababa, Ethiopia. An official letter was sent to the study sites and the study was initiated after obtaining permission from the respective health facilities. Written informed consent and/ or assent was obtained from all study participants, using the local language where appropriate.

## Study design, period and areas

A health-facility based cross-sectional study was conducted from February 2021 to September 2022. Based on the annual parasite index (API), malaria risk stratification in Ethiopia is divided into five classes: i) malaria free, ii) very low, iii) low, iv) moderate and v) high malaria risk.

The study sites were selected from different malaria risk stratification areas. The sites include: Arbaminch, Dubti, Gambella, Metehara, and Shewarobit (*Fig 1*). Arbaminch hospital is located in the Southern Nation and Nationalities People Regional State in the southern part of the country (altitude 1200 m; Lat, Long: 6.02043, 37.56788); Dubti hospital is located in the Afar Regional State in the northern-east part of the country (altitude 379 m; Lat, Long: 11.72654, 41.09440); Gambella hospital is located in Gambella Regional State in the western part of the country (altitude 447 m; Lat, Long: 8.24810, 34.59071); Metehara hospital is located in Oromia Regional State in the eastern part of the country (altitude 959 m, Lat, Long: 8.89932, 39.91726); and Shewarobit hospital is located in Amhara Regional State in the northern part of the country (altitude 1268 m; Lat, Long: 10.01734, 39.91253).

## Sample collection

Blood samples diagnosed positive for *P. vivax* (pure or mixed *P. vivax/P. falciparum*) by qPCR from patients seen at health facilities were included to determine their Duffy status. The sample size was calculated based on the proportion of Duffy-negative (20%) cases previously reported in Ethiopia [23].

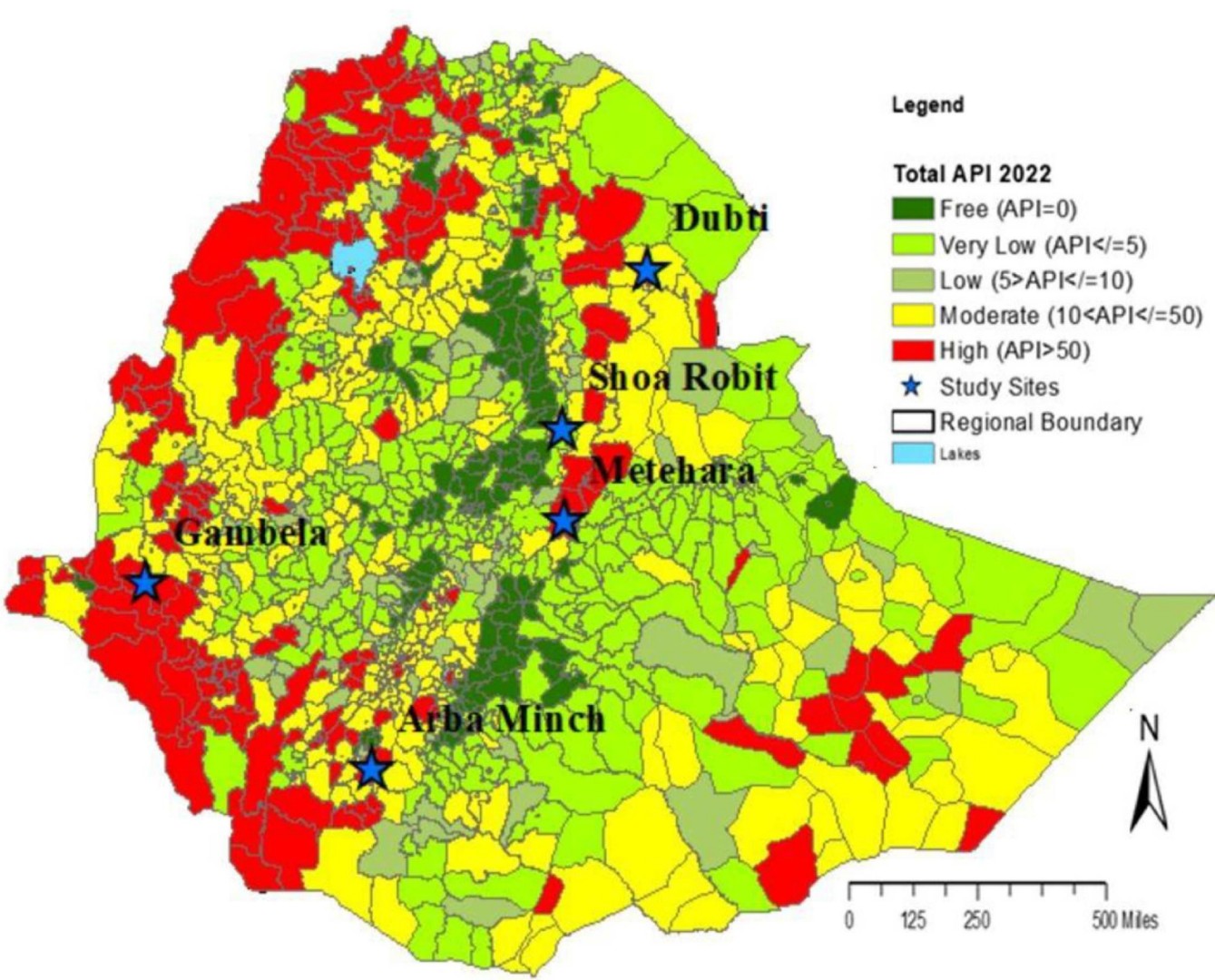

**Fig 1. A map of the study sites with malaria risk stratification background, Ethiopia.**

## Malaria diagnosis by Rapid Diagnostic Tests (RDT) and microscopy

Finger prick blood samples were collected from self-reported febrile patients seeking malaria diagnosis at a participating health centre. The malaria laboratory diagnosis was carried out for each suspected malaria patient using RDT and microscopy. SD BIOLINE Malaria Ag *Pf/Pv* (05FK80) RDT was used for the detection of HRP-II (Histidine-rich protein II) specific for *P. falciparum* and *Plasmodium* lactate dehydrogenase (pLDH) specific for *P. vivax*. Based on WHO public report, the panel detection score of the kit was 96.0% at 200 parasites/μl, and 95.0% at 200 parasites/μl for *P. falciparum*, and *P. vivax*, respectively [24]. Five microliters (5μl) of sample were obtained for the test and the result was interpreted after 15 minutes (up to 30 minutes) according to the manufacturer's guidelines.

Thick and thin blood films (stained with 10% Giemsa staining solution—pH 7.2—for 10 minutes) were prepared from finger-prick blood samples. Parasite densities were determined on thick films by expert malaria microscopists for both sexual and asexual stages of the parasites, considering a minimum of 200 WBCs, the reported number of parasites per microliter of

blood, assuming a total white blood cell count of 8000/μL. *P. vivax* sexual stages/gametocytes quantification was performed on the blood film with *P. vivax* (pure) and a mixed species (*P. vivax/P. falciparum*). Parasitemia was then classified as low (<1000 parasites per microliter of blood), intermediate (1000–4999 parasites per microliter of blood) and high (≥5000 parasites per microliter of blood) [25]. Slides were declared negative after examination of at least 100 high-power microscope fields [26]. Blood film slides were re-examined and quantified by WHO certified malaria microscopists. Patients who were positive for *P. vivax* (pure) and mixed *P. vivax/P. falciparum* infection by both RDT and microscopy were included in the study. Venous blood samples were collected and spotted onto Whatman 903 Protein Saver Card, US (Dried Blood Spot, DBS), for molecular analysis of the Duffy genotypes.

### DNA extraction and molecular diagnosis of *Plasmodium* species

Genomic DNA (human and parasite) was extracted from a 6 mm diameter spot of a DBS sample and eluted in 100μl of TE Buffer (Tris-EDTA) using the QIAamp DNA Extraction Kit (Cat.No: 79216, Lot: 172018338, Germany) according to the manufacturer's instructions for the molecular identification of malaria parasites and Duffy genotypes. The extracted DNA was stored at −20˚C for further use.

Primers previously designed to target the mitochondrial species-specific *cytochrome b (cytb)* gene were used for the detection and identification of *P. vivax* and *P. falciparum* (*Table 1*). Given the multiple copies of the mitochondria, this assay allows the detection of low density *Plasmodium* infections in malaria pre-elimination settings [27].

The PCR amplifications were carried out using 10 μL of Advanced Mix Sybgreen (2x) (Catalog # 1725271, Bio-Rad laboratories Inc. United State), 4 μL of water, 0.5 μL of each primer, and 5 μL of DNA template with the final volume of 20 μL in a Bio-Rad CFX96 Real-Time PCR detection system.

### Molecular identification of the Duffy genotypes

Molecular identification of the Duffy genotype was based on a novel sequence-specific PCR primers (Amplification Refractory Mutation System, ARMS) that allows amplifications of the GATA-1 transcription factor-binding site of the *DARC* gene promoter only when the target allele (TCT for Duffy-positive and CCT for Duffy-negative genotypes) is detected in the sample. Different sets of primers were designed and tested using Duffy-positive and Duffy-negative DNA controls which were validated by PCR and Sanger sequencing (see **S1 *Appendix***) [28] to detect the Duffy genotype by qPCR. The forward primers contain a nucleotide mismatch at the 3′ end (T for Duffy-positive and C for Duffy-negative genotypes) in addition to a mismatch base near to the SNP at the 3' end (C for Duffy-positive and A for Duffy-negative genotypes), to avoid false-amplification and prevent binding to non-complementary sequence and terminate amplification. The reverse primer are the same (*Table 2*). All Duffy negative samples found by PCR ARMS have been confirmed by Sanger sequencing.

**Table 1. Species-specific primers used for the detection and the species identification of P. vivax and P. falciparum.**

| Species | Forward or Reverse | Primer |
| --- | --- | --- |
| *P. vivax* | Forward | 5'- TGCTACAGGTGCATCTCTTGTATTC -3' |
| | Reverse | 5'- ATTTGTCCCCAAGGTAAAACG -3' |
| *P. falciparum* | Forward | 5'- ATGGATATCTGGATTGATTTTATTTATGA -3' |
| | Reverse | 5'- TCCTCCACATATCCAAATTACTGC -3' |

**Table 2. AMRS PCR primers used for the detection of Duffy-positive and -negative genotypes.**

| Duffy genotype | Forward or Reverse | Primer |
|---|---|---|
| positive | Forward | 5'- CCCTCATTAGTCCTTGGCTCTCAT -3' |
| | Reverse | 5'- CACCCTGTGCAGACAGTTCC -3' |
| negative | Forward | 5'- CCCTCATTAGTCCTTGGCTCTAAC -3' |
| | Reverse | 5'- CACCCTGTGCAGACAGTTCC -3' |

Briefly, polymorphisms in the GATA-1 were detected in two separate real-time PCR reactions. Real-time PCR reactions were performed with 10 μL of ssAdvanced mix sybgreen 2x, 10μM of primers and 5 μL of DNA extract, and amplified on the CFX96 PCR instrument (Bio-Rad, Marnes-la-Coquette, France). PCR amplification was carried out under the following conditions: heating at 98°C for 3 min, followed by 35 cycles of heating at 95°C for 20 s, and of annealing/extension at 62°C for 30 s and a final extension step at 72°C for 10 min. After completion of the amplification reaction, a melting curve (65–90°C) was generated to detect non-specific PCR amplification. For each run, homozygous (TCT/TCT), heterozygous (TCT/CCT) Duffy-positive, and homozygous Duffy-negative (CCT/CCT) samples were included as positive controls and a no template (water) as negative control (see **S2 *Appendix***).

## Statistical analyses

Data were analysed using Statistical Package for Social Sciences (SPSS, version 25). Chi-squared, or Fisher's exact tests were used for frequency data (expressed as percentages). Associations between Duffy genotypes and different variables were assessed and two-sided p-values less than 0.05 were considered as statistically significant.

## Results

### Characteristics of study participants

A total of 361 PCR-confirmed *P. vivax* (mono or mixed) infected samples were included for the molecular analysis of Duffy status. Among them, 126 (34.9%) were female, the mean age was 25 years old (ranging from 1–80 years old), 108 (29.9%) had no occupation, and 152 (42.1%) participants live in rural areas (***Table 3***).

### Distribution of the Duffy genotypes by study sites, study participants, ethnicity, gender, and age

From the 361 samples tested, 345 (95.6%) of the *P. vivax* patients were Duffy-positives (272 (78.8%) were heterozygous and 73 (21.2%) were homozygous) and 16 (4.4%) were Duffy-negatives.

Based on Duffy genotyping, 73 (20.2%) were TCT/TCT, 272 (75.4%) were TCT/CCT and 16 (4.4%) were CCT/CCT. The Duffy-negative patients (CCT/CCT genotype) from Arba-minch, Gambella, Metehara and Shewa robit was 2.7%, 8.1%, 4.0%, and 7.1%, respectively (***Fig 2***). No Duffy-negatives were detected from Dubti. The highest proportion of Duffy-negatives was in Gambella (8.1%, 6/74) and the lowest in Arbaminch (2.7%, 2/74). There was no significant association between Duffy genotype and study site (*p = 0.12*). The highest proportion of Duffy-negatives was observed in the Nuer ethnic group (12.5%, 2/16) followed by the Agnuwak group (7.7%, 1/13). No Duffy-negatives was identified among the Somale, Afar, Tigre, Welayita and Hadiya ethnic groups. No significant association was found between Duffy genotype and ethnic group (*p = 0.50*). About 5.5% (13/235) male and 2.4% (3/126)

**Table 3. Sociodemographic characteristics of study participants in Ethiopia (n = 361).**

| Variables | Category | Number | Percent (%) |
|---|---|---|---|
| Gender | Female | 126 | 34.9 |
| | Male | 235 | 65.1 |
| Age stratification in years | ≤14 | 40 | 11.1 |
| | 15–24 | 135 | 37.4 |
| | 25–34 | 122 | 33.8 |
| | ≥35 | 64 | 17.7 |
| Ethnic | Amhara | 106 | 29.4 |
| | Anuak | 13 | 3.6 |
| | Somale | 6 | 1.7 |
| | Oromo | 76 | 21.1 |
| | Afar | 52 | 14.4 |
| | Tigre | 11 | 3.0 |
| | Welayita | 30 | 8.3 |
| | Gamo | 37 | 10.2 |
| | Sidama | 11 | 3.0 |
| | Hadiya | 3 | 0.8 |
| | Nuer | 16 | 4.4 |
| Study Sites | Arbaminch | 74 | 20.5 |
| | Dubti | 68 | 18.8 |
| | Gambella | 74 | 20.5 |
| | Metehara | 75 | 20.8 |
| | Shewarobit | 70 | 19.4 |

female participants were Duffy-negatives, with no significant association between Duffy genotype and gender ($p = 0.16$). From different age stratification [22], the lowest proportion of Duffy-negatives was identified from the age groups ≤14 (2.5%, 1/40) and 25–34 (2.5%, 3/122), while the highest proportion was identified from the age group ≥35 (7.8%, 5/64), no significant association between Duffy genotype and age groups ($p = 0.337$). (Table 4). Similarly, among the Duffy-positive participants, the relative proportions of heterozygous and homozygous were not associated with either study site ($p = 0.47$), ethnic group ($p = 0.47$) or gender ($p = 0.17$) (Table 5).

## Distribution of the Duffy genotype by P. vivax infections

For the 16 *P. vivax* infections in Duffy negatives, 12 (75%) were mono-infection and four (25%) were mixed *P. vivax/P. falciparum* infections. For the 345 *P. vivax* infections in Duffy positives, 312 (90%) were mono-infections and 33 (~10%) were mixed *P. vivax/P. falciparum* infections. The proportion of mixed *P. vivax/P. falciparum* infections was two-fold higher in Duffy-negative than in Duffy positive individuals ($p = 0.047$), despite small samples for the Duffy negatives (Table 6). Of the 73 homozygous Duffy-positives, 68 (93.2%) were pure *P. vivax* infections and 5 (6.8%) were mixed *P. vivax/P. falciparum* infections, slightly lower than that observed in heterozygous Duffy-positives of which 244 (78.2%) were pure *P. vivax* infections and 28 (10.3%) were mixed *P. vivax/P. falciparum* infections (p = 0.37) (Table 6).

## Asexual and sexual parasitemia comparison by Duffy groups

Of the samples with an asexual stage density of <1,000 p/μL and ranging from 1,000 to 4,999 p/μL, 9/60 (15.0%) and 7/131 (5.3%) were obtained from Duffy-negative individuals,

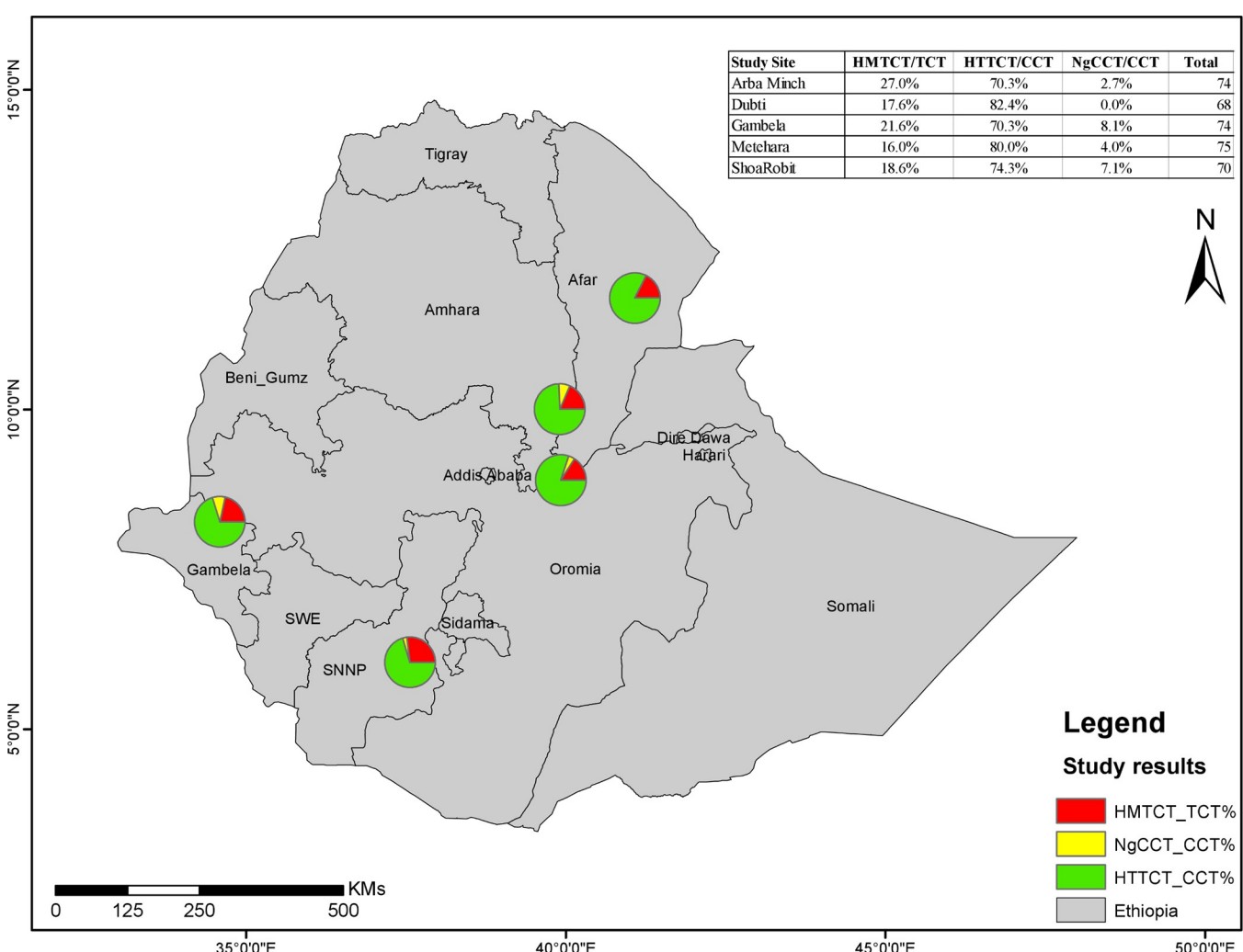

| Study Site | HMTCT/TCT | HTTCT/CCT | NgCCT/CCT | Total |
|---|---|---|---|---|
| Arba Minch | 27.0% | 70.3% | 2.7% | 74 |
| Dubti | 17.6% | 82.4% | 0.0% | 68 |
| Gambela | 21.6% | 70.3% | 8.1% | 74 |
| Metehara | 16.0% | 80.0% | 4.0% | 75 |
| ShoaRobit | 18.6% | 74.3% | 7.1% | 70 |

**Fig 2. Map showing the distribution of the Duffy genotypes at five different malaria endemic sites in Ethiopia.**

respectively. Interestingly, no Duffy-negatives were detected with asexual parasite density ≥5000 p/μL but only Duffy positive patients (n = 170; *Table 7*). The mean asexual parasite density in homozygous and heterozygous Duffy-positives was 12,165 p/μl (IQR25-75: 1,640–24,234 p/μl) and 11,655 p/μl (IQR25-75: 1,676–14,065 p/μl), respectively, significantly higher than that in Duffy-negatives (1,227p/μl; IQR25-75: 539–1,732p/μl) (*p<0.001; Fig 3*).

Of the samples with a sexual gametocytemia of <1,000 p/μL, 8/180 (4.4%) were detected in Duffy-negative individuals (*Table 7*). The other eight *P. vivax* infections in Duffy negatives were detected with no gametocytes. By contrast, 172/180 (95.6%) of the Duffy-positives were detected with sexual gametocytemia of <1,000 p/μL, 31 were detected with gametocytemia of 1,000–4999 p/μL, and 4 with gametocytemia of ≥5,000 p/μL.

## Discussion

We report here the association between 361 pure *P. vivax* and mixed *P. vivax/P. falciparum* infections collected from five eco-epidemiological study sites in Ethiopia and the Duffy genotype of the patients. We assessed malaria infection by qPCR and polymorphisms in the

**Table 4. Analysis of Duffy status by study sites (altitude), study participants ethnicity and gender (n = 361).**

| Variables | | Duffy Status | | | P-value |
|---|---|---|---|---|---|
| | | Positive n (%) | Negative n (%) | Total | |
| **Study sites (altitude)** | Arbaminch(1200 m) | 72 (97.3) | 2 (2.7) | 74 | 0.089 |
| | Dubti (379 m) | 68 (100) | 0 | 68 | |
| | Gambella (447 m) | 68 (91.9) | 6 (8.1) | 74 | |
| | Metehara (959 m) | 72 (96.0) | 3 (4.0) | 75 | |
| | Shewarobit(1268 m) | 65 (92.9) | 5 (7.1) | 70 | |
| **Ethnic of Study participant** | Amhara | 100 (94.3) | 6 (5.7) | 106 | 0.41 |
| | Agnuwak | 12 (92.3) | 1 (7.7) | 13 | |
| | Somale | 6 (100 | 0 | 6 | |
| | Oromo | 71 (93.4) | 5 (6.6) | 76 | |
| | Afar | 52 (100) | 0 | 52 | |
| | Tigre | 11 (100) | 0 | 11 | |
| | welayita | 30 (100) | 0 | 30 | |
| | Gamo | 35 (94.6) | 2 (5.4 | 37 | |
| | Sidama | 11 (100) | 0 (0) | 11 | |
| | Hadiya | 3 (100% | 0 | 3 | |
| | Nuer | 14 (87.5) | 2 (12.5) | 16 | |
| **Gender** | Female | 123 (97.6) | 3 (2.4) | 126 | 0.19 |
| | Male | 222 (94.5) | 13 (5.5) | 235 | |
| **Age in years** | ≤14 | 39 (97.5%) | 1 (2.5%) | 40 | 0.34 |
| | 15–24 | 128 (94.8%) | 7 (5.2%) | 135 | |
| | 25–34 | 119 (97.5%) | 3 (2.5%) | 122 | |
| | ≥35 | 59 (92.2%) | 5 (7.8%) | 64 | |

GATA-1 promoter affecting Duffy status by a novel ARMS-PCR approach. Here, we observed that the proportion of Duffy-negative (4.4%) among *P.vivax* infected patients was lower than previous studies conducted in different countries; Madagascar (9.3%) [28], different parts of Ethiopia (10.5–11.9%) [16], and Sudan 17.9% [9], but relatively higher than study reports such as Ethiopia 2.9% [22], south-eastern Iran 1.3% [29], Brazilian Amazon 0.6% [18]. Duffy blood groups remain highly polymorphic between different populations [30]. Historically, individuals with the Duffy-negative phenotype were thought to be resistant to *P. vivax* infection [18,19,31–33], but recent reports have shown *P. vivax* infection in Duffy-negative individuals [20,34–38] in many parts of Africa.

This study identified 4.4% (16/361) *P. vivax* infections in Duffy-negative patients. This is different from Central or West Africa, where the Duffy-negative genotype is much more common and almost all *P. vivax* infections occur among Duffy-negative individuals [37,39,40], despite relatively low number of *P. vivax* cases compared to East Africa. Several hypotheses have been put forward to explain the change in the pattern of infection in Duffy-negative individuals. One is a change in the host susceptibility, which in turn may drastically affect the distribution of *P. vivax* [9] and another is that *P. vivax* may have evolved to adapt to a wider range of hosts, environments, and highly competent transmission vectors in Africa [34]. Alternatively, *P. vivax* parasites may have developed alternative mechanisms to bind and invade reticulocytes as other host cell receptors have been shown to be involved in erythrocyte invasion [41].

Polymorphisms in Duffy blood group among different ethnic groups may contribute to the changing epidemiology of *vivax* malaria in Africa [42]. The Ethiopian population has a very

**Table 5. Analysis of heterozygous and homozygous Duffy positive patients with study site (altitude), ethnicity, and gender (n = 312).**

| Variables | | Duffy Positive | | | P-value |
|---|---|---|---|---|---|
| | | Homozygous n (%) | Heterozygous n (%) | Total | |
| **Study Sites (altitude)** | Arbaminch | 20 (27.8) | 52 (72.2) | 72 | 0.47 |
| | Dubti | 12 (17.6) | 56 (82.4) | 68 | |
| | Gambella | 16 (23.5) | 52 (76.5) | 68 | |
| | Metehara | 12 (16.7) | 60 (83.3) | 72 | |
| | Shewarobit | 13 (20.0) | 52 (80.0) | 65 | |
| **Ethnicity of study participants** | Amhara | 20 (20.0) | 80 (80.0) | 100 | 0.47 |
| | Agnuwak | 4 (33.3) | 8 (66.7) | 12 | |
| | Somale | 0 (0.0) | 6 (100) | 6 | |
| | Oromo | 13 (18.3) | 58 (81.7) | 71 | |
| | Afar | 10 (19.2) | 42 (80.8) | 52 | |
| | Tigre | 2 (18.2) | 9 (81.8) | 11 | |
| | welayita | 7 (23.3) | 23 (76.7) | 30 | |
| | Gamo | 10 (28.6) | 25 (71.4) | 35 | |
| | Sidama | 5 (45.5) | 6 (54.5) | 11 | |
| | Hadiya | 0 (0) | 3 (100) | 3 | |
| | Nuer | 2 (14.3) | 12 (85.7) | 14 | |
| **Gender** | Female | 31 (25.2) | 92 (74.8) | 123 | 0.17 |
| | Male | 42 (18.9) | 180 (81.1) | 222 | |
| **Age in years** | ≤14 | 4 (10.3%) | 35 (89.7%) | 39 | 0.284 |
| | 15–24 | 31 (24.2%) | 97 (75.8%) | 128 | |
| | 25–34 | 24 (20.2%) | 95 (79.8%) | 119 | |
| | ≥35 | 14 (23.7%) | 45 (76.3%) | 59 | |

heterogeneous ethnic composition. Eleven different ethnic groups were included in this study and the proportion of Duffy-negative and Duffy-positive varied between ethnic groups and study sites. A high proportion of Duffy-negative samples were observed among the Nuer followed by the Agnuwak ethnic groups, but no Duffy-negative was identified among the Somale, Afar, Tigre, Welayita and Hadiya ethnic groups. Other studies have also showed variation of Duffy genotype between ethnic groups. For example, in Sudan the Duffy-negative phenotype was identified in 83% of Shagia and 77% of Manasir tribes [43]; in Mauritania 54% of Moors were Duffy-positives while only 2% of black ethnic groups were Duffy-positives [42]; in Colombia Duffy genotypes were significantly associated with ethnicity [44]. These findings suggest that ethnic groups may be genetically distinct, resulting in different levels of susceptibility to *P. vivax* infection, in addition to different hypothesis such as *P. vivax* trains with a

**Table 6. Analysis of P. vivax pure and mixed P. vivax/P. falciparum infections with Duffy genotype (n = 361).**

| Variables | | *P. vivax* infections | | | Chi-Square | P-value |
|---|---|---|---|---|---|---|
| | | Pure n (%) | Mixed n (%) | Total | | |
| **Duffy genotype** | positive | 312 (90.4) | 33 (9.6) | 345 | 3.96 | 0.047 |
| | negative | 12 (75) | 4 (25) | 16 | | |
| | Total | 324 (89.8) | 37 (10.2) | 361 | | |
| **DuffyPositive** | homozygous | 68 (93.2) | 5 (6.8) | 73 | 0.79 | 0.37 |
| | heterozygous | 244 (89.7) | 28 (10.3) | 272 | | |
| | Total | 312 (90.4) | 33 (9.6) | 345 | | |

**Table 7. Association of Duffy status with asexual parasite density (n = 361).**

| Variables | | Duffy Status | | | Chi-Square | p-value |
|---|---|---|---|---|---|---|
| | | Positive n (%) | Negative n (%) | Total | | |
| Asexual stage density | <1000 | 51 (85.0) | 9 (15.0) | 60 | 23.96 | <0.001 |
| | 1000–4999 | 124 (94.7) | 7 (5.3) | 131 | | |
| | ≥5000 | 170 (100) | 0 (0) | 170 | | |

new capacity for erythrocyte invasion [28], or use several receptor-ligand interactions to tightly bind erythrocytes in the absence of a Duffy receptor [21].

Duffy-negatives with pure *P. vivax* and mixed *P. vivax/P. falciparum* infections was identified in the current study. The proportion of mixed *P. vivax/P. falciparum* infections in Duffy-negative was 25% in the present study, which was similar with the findings from Nigeria (25%)

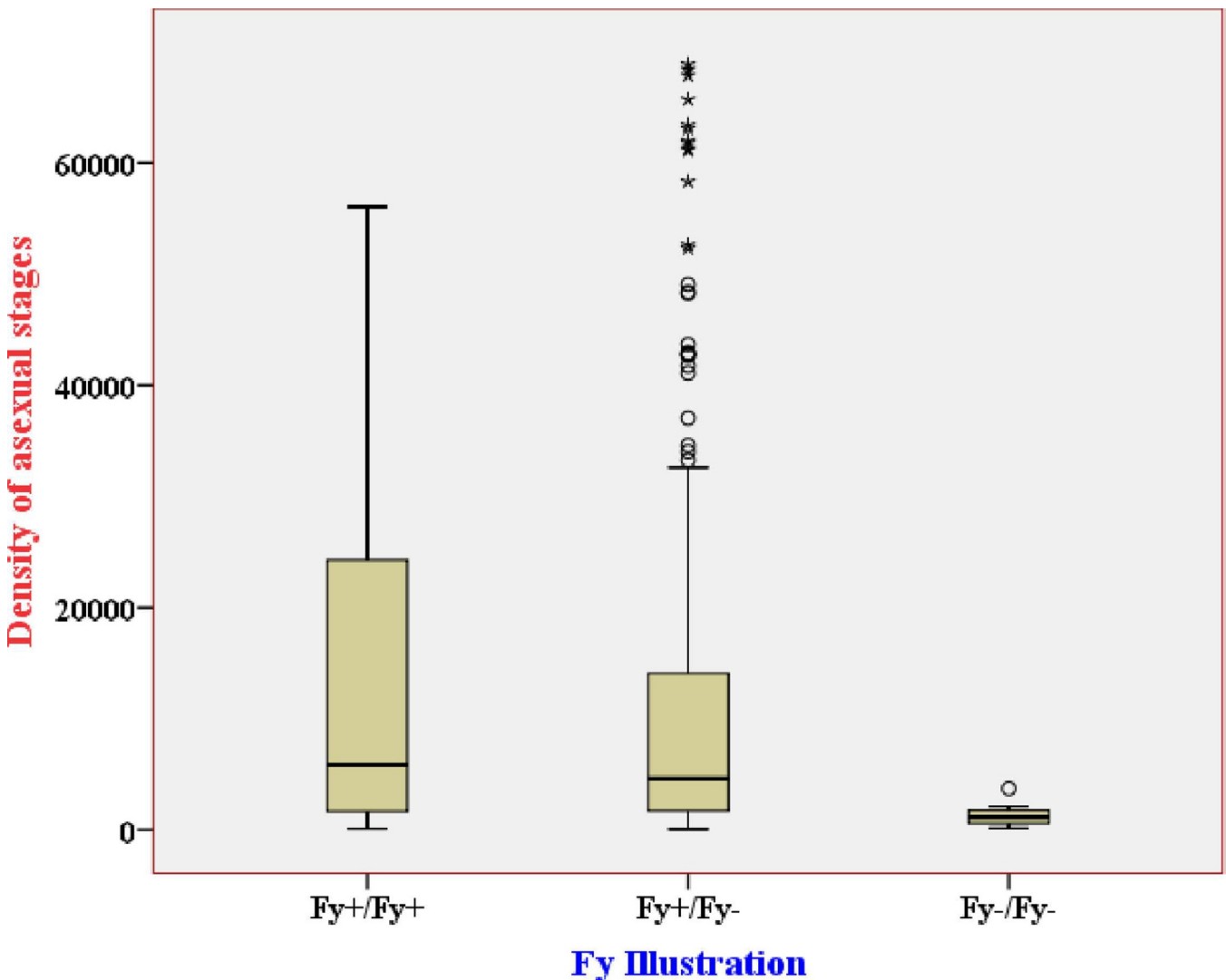

**Fig 3. Box plot showing the distribution of asexual parasite density among identified Duffy status.** Key: Fy+/+; homozygous Duffy-positive, Fy+/-; Heterozygous Duffy-positive, Fy-/-; Duffy-negative.

[45], and Cameroon (25%) [46], while it was lower than the study conducted in Madagascar (26.7%) [47], and Democratic Republic of the Congo (64%) [39]. The results provide evidence that Duffy-negative individuals can be infected by pure *P. vivax* alone or together with *P. falciparum*.

The mean asexual parasite density among Duffy-negatives (CCT/CCT) was 1,227 p/µl, around tenfold lower compared to homozygous Duffy-positives (TCT/TCT; 12,165 p/µl) and heterozygous Duffy-positive (TCT/CCT; 11,655 p/µl). Several prior studies have shown that *P. vivax* infections observed in Duffy-negatives have a significantly lower parasitemia than in Duffy-positive infections and are asymptomatic [13,16]. The low parasite densities found in Duffy-negative malaria patients may indicate that *P. vivax* uses an alternative pathway to invade Duffy-negative reticulocytes, which can be much less efficient than the classical pathway involving the interaction between PvDBP and *DARC* [22]. Alternatively, it has been hypothesized that the reticulocytes of Duffy-negative individuals may express the *DARC* protein in a transient and weak manner, thereby mediating a low invasion capacity [16]. Among the 16 Duffy negative samples detected in this study, half were detected with low-level gametocytes. The presence of gametocytes among Duffy negatives may further imply that these infections have the potential contributing to onward transmission.

To conclude, this study reports heterogeneous proportions of Duffy-negative individuals among *P. vivax* malaria patients across Ethiopia, confirms that Duffy-negativity does not provide complete protection against *P. vivax* infection, and these infections are often associated with low parasitemia, which may represent hidden reservoirs that can contribute to transmission. Therefore, the development of *P. vivax*-specific elimination strategies, including alternative antimalarial vaccines other than a Duffy-binding protein-based vaccine, should be facilitated by a better understanding of the relationship between Duffy-negativity and the invasion mechanisms of *P. vivax*.

## Supporting information

**S1 Appendix. Figures summarizing Sanger sequencing data.**
(TIF)

**S2 Appendix. Figures summarizing AMRS PCR-based allelic discrimination amplification curves in a control samples validated by PCR/Sanger sequencing.**
(TIF)

## Acknowledgments

We would like to thank i) SIDA project, ii) Human, Heredity and Health in Africa (H3Africa) [H3A-18-002]. H3Africa is managed by the Science for Africa Foundation (SFA Foundation) in partnership with Wellcome, NIH and AfSHG and iii) the Institute Pasteur, Paris, the University of Strasbourg, and the French Government (Agence Nationale de la Recherche; ANR-18-CE15-0026, ANR-21-CE35-0006 and the Laboratoire d'Excellence 'French Parasitology Alliance for Health Care', ANR-11-15 LABX-0024-PARAFRAP), and iv) EPHI for the materials support. We would also like to thank all the individuals who contributed technically to this study: the laboratory staff at the study sites, the staff at ALIPB (parasitology department), the staff at the Institute of Parasitology and Tropical Diseases, University of Strasbourg (Cécile Doderer-Lang), and those who were provided support on Arc Map and statistical analysis (Dereje Dilu, Abriham Keraleme and Chalie). We must express our sincere gratitude to all the regional health offices and the study sites for their cooperation. Last but not least, our sincere thanks go to all the participants who took part in this study.

## Author Contributions

**Conceptualization:** Abnet Abebe.

**Data curation:** Abnet Abebe, Sisay Dugassa, Ashenafi Assefa, Jonathan J. Juliano, Eugenia Lo, Didier Menard, Lemu Golassa.

**Formal analysis:** Abnet Abebe, Isabelle Bouyssou, Solenne Mabilotte, Sisay Dugassa, Ashenafi Assefa, Jonathan J. Juliano, Eugenia Lo, Didier Menard, Lemu Golassa.

**Funding acquisition:** Abnet Abebe.

**Investigation:** Abnet Abebe, Didier Menard, Lemu Golassa.

**Methodology:** Abnet Abebe, Eugenia Lo, Didier Menard, Lemu Golassa.

**Project administration:** Abnet Abebe.

**Resources:** Abnet Abebe, Didier Menard, Lemu Golassa.

**Software:** Abnet Abebe.

**Supervision:** Abnet Abebe, Sisay Dugassa, Ashenafi Assefa, Jonathan J. Juliano, Eugenia Lo, Didier Menard, Lemu Golassa.

**Validation:** Abnet Abebe, Eugenia Lo, Didier Menard, Lemu Golassa.

**Visualization:** Abnet Abebe, Lemu Golassa.

**Writing – original draft:** Abnet Abebe.

**Writing – review & editing:** Abnet Abebe, Sisay Dugassa, Ashenafi Assefa, Jonathan J. Juliano, Eugenia Lo, Didier Menard, Lemu Golassa.

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
