## [Decision Letter · Decision Letter 0]

25 May 2023

Dear Mr Abebe,

Thank you very much for submitting your manuscript "Potential hidden Plasmodium vivax malaria reservoirs from low parasitemia Duffy-negative Ethiopians: molecular evidence" for consideration at PLOS Neglected Tropical Diseases. As with all papers reviewed by the journal, your manuscript was reviewed by members of the editorial board and by several independent reviewers. The reviewers appreciated the attention to an important topic. Based on the reviews, we are likely to accept this manuscript for publication, providing that you modify the manuscript according to the review recommendations (please see suggestion by reviewer #3). 

Sincerely,

Kamala Thriemer

Academic Editor

Charles Jaffe

Section Editor

Reviewer's Responses to Questions

**Key Review Criteria Required for Acceptance?**

**Methods**

-Are the objectives of the study clearly articulated with a clear testable hypothesis stated?

-Is the study design appropriate to address the stated objectives?

-Is the population clearly described and appropriate for the hypothesis being tested?

-Is the sample size sufficient to ensure adequate power to address the hypothesis being tested?

-Were correct statistical analysis used to support conclusions?

-Are there concerns about ethical or regulatory requirements being met?

Reviewer #1: The objectives of the study are well defined, the methodology and study design appropriate for the objectives and results from the analysis support the conclusions.

Reviewer #2: -Are the objectives of the study clearly articulated with a clear testable hypothesis stated?

Yes. The aim of this study was to assess the Duffy status of 50 patients with P. vivax infection from different study sites in Ethiopia. Although several studies have demonstrated the presence of P. vivax infection in Duffy-negative individuals, the distribution of these infections in different eco-epidemiological zones and ethnic groups are not well documented, especially in Ethiopia. This study compares the prevalence of Duffy blood group genotypes of P. vivax infected patients among different eco-epidemiological zones of Ethiopia.

-Is the study design appropriate to address the stated objectives?

Yes, a health-facility based cross-sectional study was used to recruit patients from areas with P. vivax as either mono-infections or co-infections with P. falciparum from five representative sites from areas with different transmission levels. Diagnosis was performed by RDT, microscopy, and qPCR to ensure accurate diagnosis. Blood samples were collected on Whatman 903 Protein Saver Cards, the standard for dried blood spot samples intended for use in molecular studies. DNA extractoin was performed using a QIAamp DNA Extraction Kit, which performs well with dried blood spot samples. A novel PCR was used to identify the Duffy genotype, but all Duffy negative samples were confirmed by Sanger sequencing.

-Is the population clearly described and appropriate for the hypothesis being tested?

Yes, table 3 contains all relevant demographic data for the population. The female to male ratio is acceptable and the age range of the population is appropriate for the population at risk of severe disease. The authors also took care to ensure that many different ethnic backgrounds were appropriately represented in the study population from among the five study sites.

-Is the sample size sufficient to ensure adequate power to address the hypothesis being tested?

Yes, 361 PCR-positive P. vivax-infected participants were enrolled in the study.

-Were correct statistical analysis used to support conclusions?

Yes, SPSS was used to perform Chi-squared and Fisher’s exact tests to determine the differences between Duffy homozygous, heterozygous, and negative groups. P-values less than 0,05 were used as the criteria for defining statistical signficance, the norm in the field.

-Are there concerns about ethical or regulatory requirements being met?

No, the study led by Abebe and Golassa includes researchers from the country where the study was conducted and took care to ensure that study enrollment reprsented Ethopia’s ethnically diverse population. An Ethical Statement was included in the Methods which included the ethical approval from the Institutional Review Board of the the Aklilu Lema Institute of Pathology at Addis Ababa University, with which the authors are affiliated.

Reviewer #3: The methodology used to assess the Duffy prevalence is sufficient and repeatable. Some minor comments:

* Figure 1 is challenging to read with the blue text, recommend changing color or using a different legend.

* Recommend listing the accuracy of the SD BIOLINE RDT in the methodologies to justify usage.

**Results**

-Does the analysis presented match the analysis plan?

-Are the results clearly and completely presented?

-Are the figures (Tables, Images) of sufficient quality for clarity?

Reviewer #1: Figures, tables are clear and represent well the obtained results

Reviewer #2: -Does the analysis presented match the analysis plan?

Yes, the data includes Duffy genotype, genotype by region, association between genotype and ethnic group, study site/altitude, and gender. The authors also presented Duffy positivitiy among patients with P. vivax mono- vs co-infections, and the parasite density and sexual forms by Duffy groups. 

-Are the results clearly and completely presented?

Yes, the text of the manuscript is very well written and easy to follow.

-Are the figures (Tables, Images) of sufficient quality for clarity?

Yes, figures and tables are well designed and easy to read and interpret.

Reviewer #3: All results match the analysis plan and clearly presented. However, figure 2 seems redundant with similar information being shown in figure 3, but with more geographical details. Recommend either removal of figure 2 or merging the two as a two panel figure.

The discussion section mentions the proportion of Duffy-negative infections being lower than several other countries. Is it possible to add another map showing the reported frequencies?

**Conclusions**

-Are the conclusions supported by the data presented?

-Are the limitations of analysis clearly described?

-Do the authors discuss how these data can be helpful to advance our understanding of the topic under study?

-Is public health relevance addressed?

Reviewer #1: The conclusions are supported by the results.

Reviewer #2: -Are the conclusions supported by the data presented?

Yes. Results show that Duffy-negativity is not significantly associated with any particular study site/altitude, ethnicity, gender, or age. Interestingly, mixed P. vivax/P. falciparum co-infections were two fold higher in Duffy-negative participants, making this finding statistically significant. Assessment of asexual parasite densities among Duffy positive and negative participants revealed that no Duffy-negative participants had parasite densities above 5000 parasites per microlitre while the average asexual parasite density for Duffy-positives was ~12,000 p/uL. Assessment of sexual parasite densities revealed lower levels of sexual gametocytemia among Duffy-negative participants. 

Based on this data, the authors conclude that Duffy-negativity does not provide completed protection from P. vivax infection. The authors also compared the prevalence of Duffy-negative infections to similar previous studies in Ethiopia and worldwide. The authors also compare the proportion of mixed infections among Duffy-negative individuals to previous studies. 

The authors put forward different explantions from other published articles that could explain the reason for the change in patter of infection in Duffy-negative individuals.The authors also put forward several possible explainations for the lower asexual and sexual parasite densities observe in Duffy-negative individuals from previous literature and the implications that although the sexual parasite density of Duffy-negative individuals is lower than for Duffy-positive, that the presence of gametocytes suggests that these infections have the potential to contribute to further P. vivax transmission, and that this may represent a hidden reservoir for P. vivax that can contribute to transmission. The authors advocate for improved vaccine design and understanding of the mechanisms of P. vivax invasion.

-Are the limitations of analysis clearly described?

Yes, the study is well described to indicate how the data should be interpreted based on the study population enrolled and methods used to assess the objectives. 

-Do the authors discuss how these data can be helpful to advance our understanding of the topic under study?

Yes, the authors clearly state the implications of the study including that the presence of gametocytes ins lower density P. vivax infections suggests that this population may represent a hidden reservoir for P. vivax that can contribute to transmission. 

-Is public health relevance addressed?

Yes. The authors advocate for improved vaccine design and understanding of the mechanisms of P. vivax invasion.

Reviewer #3: Conclusions were concise and addressed molecular and geographical results obtained in the study, although more explanation for alternative invasion mechanisms (lines 340-342) and elaborati

**Editorial and Data Presentation Modifications?**

Reviewer #1: (No Response)

Reviewer #2: Accept. No revisions necessary.

Reviewer #3: (No Response)

**Summary and General Comments**

Reviewer #1: This is a descriptive study on the proportion of P. vivax infected Duffy negative individuals among individuals infected with P. vivax in different regions of Ethiopia. The study presents new data in Ethiopia and eventhough the topic is not novel, it adds epidemiological data that confirms infections in Duffy negative individuals, questions the role of these infections as human reservoirs and supports the existence of alternative pathways of P. vivax invasion.

Reviewer #2: The study is well designed, figures are easy to understand, and results and conclusions are clearly stated. No revisions are necessary.

Reviewer #3: This study sought to examine the prevalence of Duffy-negative P. vivax infections over the course of 1.5 years. Abebe and colleagues did an excellent job summarizing the findings of this study and communicating the current status of an emerging health problem in Ethiopia.

PLOS authors have the option to publish the peer review history of their article (what does this mean?). If published, this will include your full peer review and any attached files.

Reviewer #1: No

Reviewer #2: Yes: Jessica N. McCaffery

Reviewer #3: No

Figure Files:

Data Requirements:

Reproducibility:

References

---

## [Editor Report · Decision Letter 1]

14 Jun 2023

Dear Mr Abebe,

We are pleased to inform you that your manuscript 'Potential hidden Plasmodium vivax malaria reservoirs from low parasitemia Duffy-negative Ethiopians: molecular evidence' has been provisionally accepted for publication in PLOS Neglected Tropical Diseases.

Best regards,

Kamala Thriemer

Academic Editor

Charles Jaffe

Section Editor

---

## [Editor Report · Acceptance letter]

26 Jun 2023

Dear Mr Abebe,

We are delighted to inform you that your manuscript, "Potential hidden Plasmodium vivax malaria reservoirs from low parasitemia Duffy-negative Ethiopians: molecular evidence," has been formally accepted for publication in PLOS Neglected Tropical Diseases.

Best regards,

Shaden Kamhawi

co-Editor-in-Chief

Paul Brindley

co-Editor-in-Chief
